



# Interannual variability of winds in the Antarctic mesosphere and lower thermosphere over Rothera (67°S, 68°W) in radar observations and WACCM-X

Phoebe E. Noble [1,2], Neil P. Hindley [1], Corwin J. Wright [1], Chihoko Cullens [3], Scott England [4], Nicholas Pedatella [5], Nicholas J. Mitchell [1,2], and Tracy Moffat-Griffin [2]

[1]Centre for Space, Atmospheric and Oceanic Sciences, Department of Electronic Engineering, University of Bath, Bath, UK
[2]Atmosphere, Ice and Climate Team, British Antarctic Survey, Cambridge, UK
[3]Laboratory for Atmospheric and Space Physics, University of Colorado, Boulder, CO, USA
[4]Virginia Polytechnic Institute and State University, Blacksburg, VA, USA
[5]High Altitude Observatory, National Center for Atmospheric Research, Boulder, CO, USA

**Correspondence:** Phoebe Noble, pn399@bath.ac.uk

**Abstract.** The mesosphere and lower thermosphere (MLT), at heights of 80-100 km, is critical in the coupling of the middle and upper atmosphere and controls the momentum and energy transfer between these two regions. However, despite its importance, many General Circulation Models (GCMs) do not extend upwards into the MLT and those that do remain poorly constrained. In this study, we use a long-term meteor radar wind dataset from Rothera (67°S, 68°W) on the Antarctic Peninsula to test

the Whole Atmosphere Community Climate Model with thermosphere-ionosphere eXtension (WACCM-X). This radar has an interferometer to determine meteor heights and has been running since 2005. This unique combination yields a dataset ideally suited to investigate interannual variability. We find that although some characteristic features in monthly median winds are represented well in WACCM-X, the model exhibits significant biases. In particular, the observations reveal a $\sim$10 ms$^{-1}$ eastward wind at heights of 85-100 km in Antarctic winter, whereas the model predicts winds of the same magnitude but of

opposite direction. We propose that this bias exists because WACCM-X is missing eastward momentum forcing in the MLT from the breaking of secondary gravity waves.

Both the model and observations reveal significant interannual variability in monthly median winds. We investigate the role of particular key external phenomena in driving the winds in this region. These phenomena are; i) variations in Solar activity, ii) the El Niño Southern Oscillation (ENSO), iii) the Quasi-Biennial Oscillation (QBO) and iv) the Southern Annular Mode

(SAM). We use a linear regression method to investigate how the observed and modelled winds, and modelled gravity wave tendencies in the Antarctic MLT vary in relation to the indices that quantify these phenomena.

We find that there are some times of year and some height ranges at which there are significant correlations between the indices and the observed/modelled winds. In particular, in summer, there is a strong positive correlation in the modelled and observed zonal winds with the 11-year Solar cycle of magnitude up to 9 ms$^{-1}$ per 70 Solar flux units. However, there appears

to be little significant influence of the ENSO on the winds observed by the radar although WACCM-X zonal winds display a negative correlation throughout January-February and a positive correlation during March-May. Results from the QBO indices are varied and we find differing correlations in the model and observations. Finally, we find a positive correlation between





observed summertime zonal winds and the SAM which has a magnitude of $9\,\mathrm{ms^{-1}}$ per 2.5 hPa change in the SAM index. However, in WACCM-X zonal winds the summertime response is negative and around $10\,\mathrm{ms^{-1}}$ per 2.5 hPa. The significance
of this work lies in our quantifying the biases in a leading GCM and demonstrating there is significant interannual variability in both modelled and observed winds, some of which are consistent with the proposal of external forcing.



## 1   Introduction

The winds in the mesosphere and lower thermosphere (MLT) are strongly influenced by gravity wave driving. Gravity waves generated at lower heights propagate upwards into the MLT, growing in amplitude before breaking and depositing their mo-
mentum and thus driving the winds (Smith, 2012).

This process poses unique problems for models of the planetary scale circulation because the details of gravity wave driving remain poorly constrained, largely due to limited available measurements of the wind in the MLT region. This limitation is particularly so in Antarctica where instrument deployment is logistically difficult. However, the Antarctic Peninsula and nearby Southern Andes mountain ranges are one of the largest gravity wave source regions. This makes observations here vital
in understanding the gravity wave driven dynamics and constrain models in this region.

The general circulation of the MLT is known to display a clear seasonal cycle. The meridional wind of the MLT is dominated by the global pole-to-pole circulation of the wave driven upper branch of the Brewer-Dobson circulation. Further, there is good evidence that the winds of the general circulation include significant interannual variability (e.g. Baumgaertner et al., 2005; Dowdy et al., 2007; Sandford et al., 2010) and a number of studies have attempted to quantify and explain the physical
causes of this variability. However, such studies are often limited by the relatively small number of reliable observational datasets available of long enough duration to be used to investigate this inter-annual variability. Despite this difficulty some observational studies have explored the possible influence of the Solar cycle on the winds in the MLT region (e.g. Greisiger et al., 1987; Bremer et al., 1997; Jacobi et al., 1997; Middleton et al., 2002; Wilhelm et al., 2019; Cai et al., 2021). As noted by Cai et al. (2021), results from such studies vary by location and time period used. Further there is no definite agreed
mechanism for Solar modulation of MLT winds. Similarly, other atmospheric phenomena have been proposed to affect the winds in the MLT with varying significance. Observational evidence of an influence from the El Niño Southern Oscillation (ENSO) has been suggested by Li et al. (2016), Llamedo et al. (2009), Sundararajan (2020) and Kishore et al. (2014) for the stratosphere/mesosphere. Some studies (Ford et al., 2009; Kishore et al., 2014) found an influence from the Quasi-Biennial Oscillation (QBO) on MLT winds. However, Baumgaertner et al. (2005) found no such influence from the Solar cycle, ENSO
or the QBO on MF radar observed Antarctic winds. In addition, the influence of the Southern Annular Mode (SAM) on MLT winds has not been widely explored aside from briefly in Merzlyakov et al. (2009) where no link was found.

Modelling studies have also investigated the interannual variability of the MLT and attempted to identify the causal drivers of such variability. In particular, Cullens et al. (2016) explored Solar cycle influences in WACCM (version 4). Gan et al. (2017) and Ramesh et al. (2020) used a linear regression method to investigate the interannual variability of winds and temperatures
in the global climate models eCMAM and WACCM, respectively. This method was used to attribute changes in the winds and temperatures to different external drivers. Both Gan et al. (2017) and Ramesh et al. (2020) found a positive temperature response to higher levels of Solar irradiance for all latitudes in the mid stratosphere and upwards, with the strength of the correlation increasing with height. As well as this temperature response, both Gan et al. (2017) and Ramesh et al. (2020) found changes in the wind correlating to the Solar cycle, although these were more varied than the temperature responses and were
latitude and height dependent.



In this work we present the first long-term study (i.e. spanning over a Solar cycle) of the interannual variability of Antarctic winds from a meteor radar equipped with height resolving capabilities and compare these observations to the predictions of WACCM-X. Long-term Antarctic MLT winds have been explored in the past by meteor radars without height resolving (e.g. Portnyagin et al., 1992) and Merzlyakov et al. (2009). Whilst some of these studies span a longer time period than that considered here, the MLT winds have strong variation with height which could not be addressed. Other long-term work has used observations from MF radars (e.g. Baumgaertner et al., 2005; Dowdy et al., 2007; Iimura et al., 2011; Portnyagin et al., 1992; Merzlyakov et al., 2009). However, MF radars have known and significant biases in winds measured at heights above ∼90 km (Manson et al., 2004; Jacobi et al., 2009; Wilhelm et al., 2017). Here we investigate data recorded from 2005-2020 by the Rothera meteor radar and build upon work done earlier by Sandford et al. (2010) who reported first results from this radar and compared to winds in Esrange (68°N, 21°E). Cullens et al. (2016) also inspires the basis for this study wherein they used WACCM to explore the influence of the 11-year Solar cycle on atmospheric winds globally. They found that in the southern hemisphere there are statistically significant changes in gravity wave drag and associated winds that are likely due to the Solar cycle. Our goal here is to determine the interannual variability found in this long radar dataset and compare the winds to the WACCM-X model. We use a linear regression method to explore the relationship between both observed and modelled MLT winds and modelled gravity wave tendencies with four particular potential external phenomena, namely, i) the Solar cycle, ii) the ENSO, iii) the QBO and iv) the SAM.

In Section 2 we present the data used; meteor radar, WACCM-X and climate indices for the linear regression. Section 3 describes the linear regression method used to explore the interannual variability. The results are split over two sections with Section 4 presenting and comparing the wind climatologies from both the radar observations and WACCM-X model. Section 5 presents the results from the linear regression, and Section 6 the gravity wave tendencies from the model. The discussion can be found in Section 7 and conclusions in Section 8.

## 2  Data

### 2.1  Meteor Radar

In this study we use wind data from the SKiYMET all sky radar at Rothera on the Antarctic Peninsula (67°S, 68°W). The radar was installed in February 2005. It uses a peak power of 6 kW and operates with a radio frequency of 32.5 MHz. Further details on the configuration of this particular radar can be found in Sandford et al. (2010) and a complete description of meteor radar processing methods in Hocking et al. (2001). Here we derive hourly zonal and meridional wind values from meteor echoes for heights of 80-100 km. A collection of meteor echoes is needed to determine the horizontal wind. We use a Gaussian weighting in height and time with a full width half maxima of 2 hrs in time and 3 km in height. This Gaussian weighting is then stepped by 1 hour in time and 1 km in height. For a complete description of this method see Hindley et al. (2022). Radio interference reduced data quality in the interval December 2009 to January 2010 and damage to the antennas reduced data quality in the interval January 2016 and December 2018. Additionally, December 2010 experienced interference from an unknown source (possibly summertime base activities at Rothera), hence we discard data from these intervals.



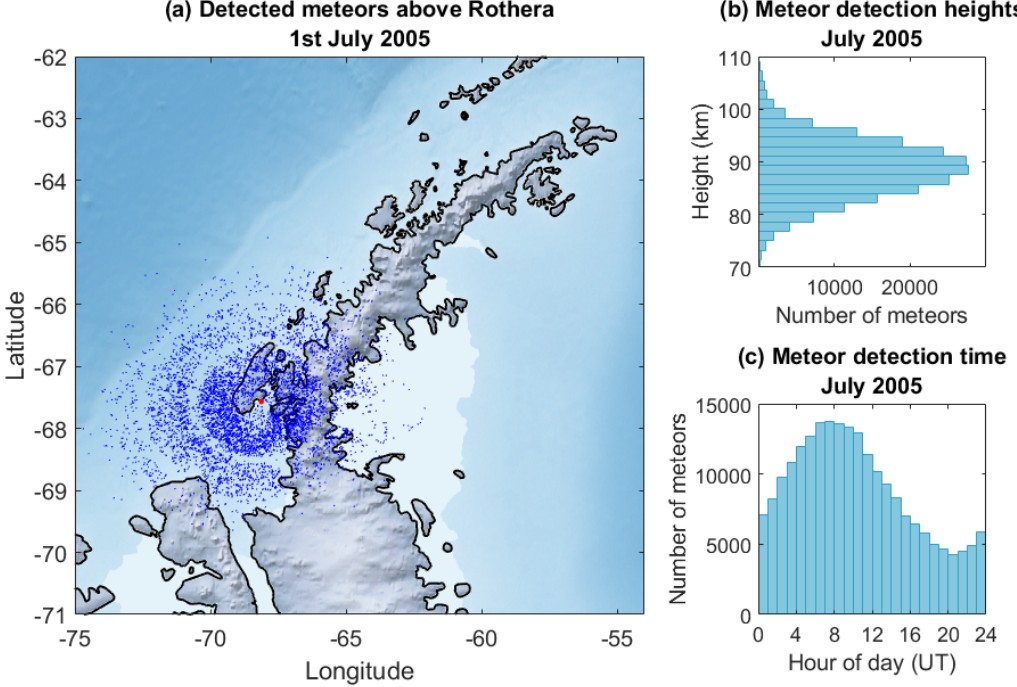

**Figure 1.** (a) The spatial distribution of meteors recorded over Rothera on one day, 1st July 2005. Individual meteors are shown in blue and the radar location in red (b) Height distribution of meteors recorded over one month, July 2005 (c) Histogram of hourly meteor counts recorded over one month, July 2005

In this section we discuss the distribution of meteors detected by the meteor radar as this can influence the calculated winds.

95 Figure 1a shows a top down view of the locations of detected meteors, 1b the distribution of the heights of the detected meteors. There is also a diurnal cycle in meteor detection which can be seen in panel c. More meteors are observed in the morning than the afternoon due to meteors being "swept" up on the leading edge of the Earth's path round the Sun. Within our wind calculation, we use a minimum threshold of 20 meteors per Gaussian window in time and height to ensure that a valid wind value is found. This means that sometimes, particularly above 100 km and below 80 km in height in the afternoon, there

100 are not enough meteors to calculate a wind value. To ensure this does not bias our averaged winds, we calculate composite days of zonal and meridional winds for each month to compensate for the diurnal cycle. From the composite days we calculate a monthly median wind value in the zonal and meridional components, in order to remove the tides and planetary waves and reveal the background winds. These monthly median winds are used throughout the rest of this study.



## 2.2 WACCM-X

The Whole Atmosphere Community Climate Model (WACCM) is a General Circulation Model (GCM) which simulates the
atmosphere from the surface to the lower thermosphere (∼140 km). WACCM-eXtended (WACCM-X), reaches to the upper
thermosphere and includes additional thermodynamic processes and ionospheric electrodynamics (Liu et al., 2018). The nu-
merical framework for WACCM-X is based on the Community Earth System Model (CESM). WACCM-X includes chemical,
dynamical and physical processes to model the lower, middle, and upper atmospheres (Neale et al., 2013; Marsh et al., 2013).
Gravity waves are modelled using a parametrisation based on Lindzen (1981) and Richter et al. (2010). The model resolution
is 2.5° latitude by 1.9° longitude.

We use data from the specified dynamics version of WACCM-X version 2.1 extended runs over the period 1980-2017 which
nudge to Modern-Era Retrospective analysis for Research and Applications, Version 2 (MERRA-2) data from the surface up
to heights of ∼50 km (Gasperini, 2019a). The details and validation of WACCM-X 2.0 can be found in Liu et al. (2018). For
direct comparability with the meteor radar data we extract data from WACCM-X as follows. Firstly, the meteor radar measures
winds over a horizontal collecting region of several hundred km diameter (as shown in Figure 1a). To enable comparison with
the model, we average the WACCM-X winds over all grid points that lie within this region. Secondly, as a height coordinate, we
take the geopotential height from WACCM-X, convert to geometric height and interpolate onto the meteor radar height grid. We
again take monthly median wind values in the zonal and meridional components for comparison with the radar observations. We
also explore the zonal gravity-wave tendencies (gravity wave drag in ms$^{-1}$ per day) from WACCM-X as gravity wave driving
strongly influences the winds. These zonal gravity-wave tendencies in WACCM-X were found to be noisy when examined over
the meteor collecting region. We therefore calculated tendencies as zonal-means in a band of 300 km latitudinal width, centred
over the latitude of Rothera.

## 2.3 Climate Indices

To explore potential drivers of interannual variability in the MLT winds, we regress our monthly median winds against a
number of climate indices, specifically the 11 year Solar cycle, the El Niño Southern Oscillation (ENSO), the Quasi-Biennial
Oscillation (QBO) and the Southern Annular Mode (SAM, also known as the Antarctic Oscillation). These phenomena are
summarised in Table 1 and presented in Figure 2.

These indices (except for SAM) are chosen as previous work has found some link between the phenomena and the MLT. For
Solar Greisiger et al. (1987); Bremer et al. (1997); Jacobi et al. (1997); Middleton et al. (2002); Cullens et al. (2016); Wilhelm
et al. (2017); Ramesh et al. (2020) and Cai et al. (2021), ENSO Llamedo et al. (2009); Li et al. (2016) and Ramesh et al. (2020)
and QBO (Ford et al., 2009; Ramesh et al., 2020). To represent the QBO we use two indices, QBO10 and QBO30, a measure
of the equatorial winds at two different pressure heights (10 hPa and 30 hPa). These two heights are chosen because they are
almost orthogonal to each other, in order to capture possible QBO responses (Chiodo et al., 2014). The influence of the SAM
is often excluded from GCM linear regression studies for the simple reason that such studies apply linear regression to a global
model where regional oscillations are not considered. However, in this study on Antarctic winds the SAM becomes relevant.

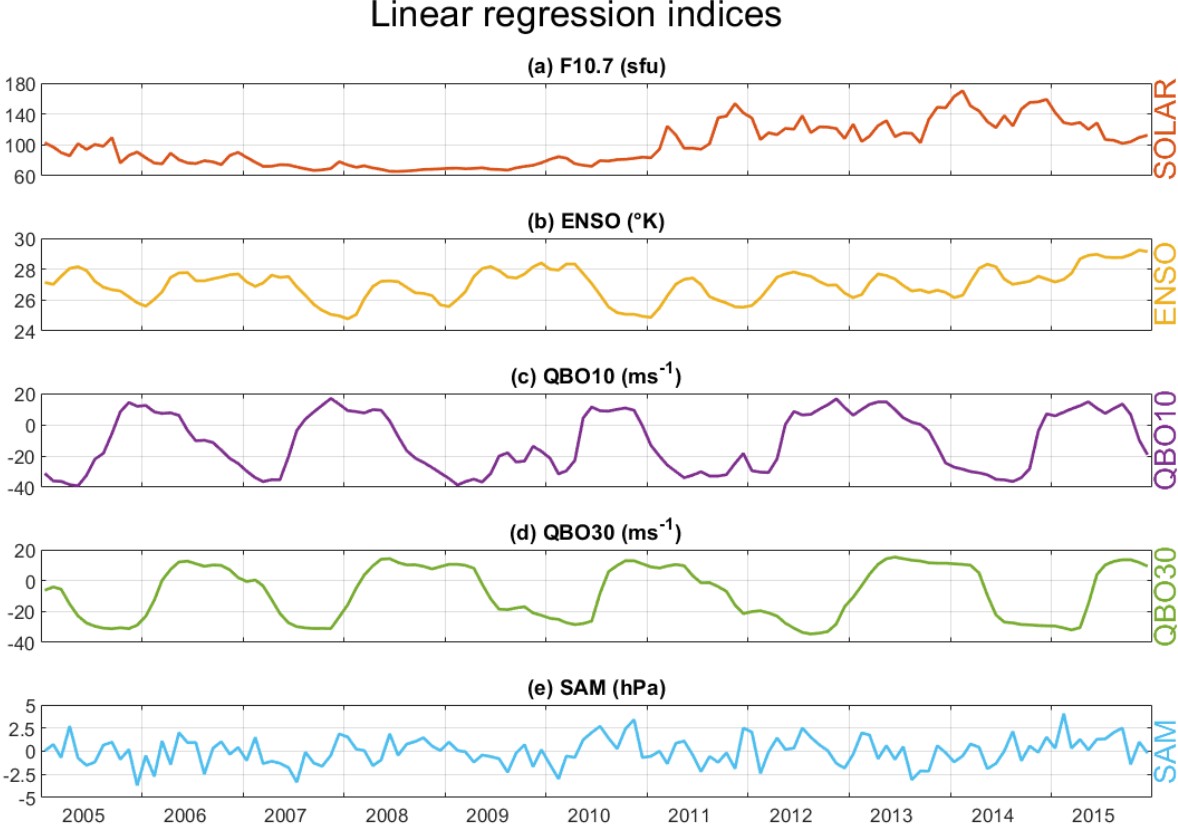

**Figure 2.** Time series of the climatological indices used in our linear regression analysis from 2005-2015.

| Climatological Index | Interdecile range ($\alpha$) | Data source |
|---|---|---|
| Solar | 70 sfu | National Research Council of Canada |
| ENSO | 4 K | Trenberth (2020) |
| QBO10 | 40 ms$^{-1}$ | ERA5 zonal-mean zonal-winds between 5°N and 5°S at 10 hPa height |
| QBO30 | 38 ms$^{-1}$ | ERA5 zonal-mean zonal-winds between 5°N and 5°S at 30 hPa height |
| SAM | 2.5 hPa | Marshall (2018) |

**Table 1.** Summary of climatological indices used in this study





## 3 Method

### 3.1 Linear Regression Analysis

To carry out our linear regression we use monthly median winds in the zonal and meridional components from the temporally
overlapping period of the radar observations and WACCM-X simulation, 2005-2015. To start, we apply a similar method to
(Gan et al., 2017; Ramesh et al., 2020) by considering the wind anomaly for each month. This wind anomaly is defined as the
deviation of each month from the climatological mean. This removes the seasonal cycle revealing only interannual variability.

We next separate our data and build a multilinear regression model for each month using a three month window centered on
the month of interest. This gives 33 datapoints for each regression and, with our 5 independent variables, allows for 27 degrees
of freedom. The linear regression analysis is carried out on each height level for the meridional and zonal winds independently.
We propose and apply the following linear regression model and use the Ordinary Least Squares (OLS) method to estimate the
unknown parameters.

$$U' = \beta_0 + \beta_1 \times \text{Solar} + \beta_2 \times \text{ENSO} + \beta_3 \times \text{QBO10} + \beta_4 \times \text{QBO30} + \beta_5 \times \text{SAM}, \tag{1}$$

where $U'$ is the wind anomaly (for radar or WACCM-X winds, zonal or meridional wind component) and Solar, ENSO,
QBO10, QBO30 and SAM are the climate indices described in Section 2.3. Statistical significance of each regression coefficient
is determined using Student's T-test. We also apply this identical linear regression method to the zonal mean gravity wave
tendencies from WACCM-X (which have units ms$^{-1}$ per day).

Linear regression is a powerful tool for identifying relationships between variables and it allows us to decompose the wind
anomaly into component parts that we may attribute to various external drivers. However, we note that correlation does not
necessarily mean causation and so a correlation between the winds and the indices could be a coincidence and not a causal
link. Further, the application requires some care in the interpretation of the results. Firstly, there are likely to be other causes
of interannual variability aside from the five indices that we regress against, but it is impossible to include everything without
over overfitting our model. Secondly, linear regression is by definition, linear, but the atmosphere is not in general a linear
system. As such, any results are potentially a simplification of the interactions that are occurring. However, they still allow us
to investigate the linear influences of atmospheric and Solar oscillations on the winds in the Antarctic MLT and gain useful
insights into potential drivers of variability.

### 3.1.1 Multicollinearity

To ensure the regression analysis is valid, we must check that there is no multicollinearity within our independent variables
i.e. none of the independent variables are correlated with each other. Correlated independent variables would lead to biased
coefficients and standard errors in the regression. To check for this, we consider the variance inflation factors (VIFs). This
method is chosen over the possible alternative of evaluating pairwise correlations as it is possible that individual pairs may
not be correlated with each other, but two or more variables together could cause multicollinearity problems with a remaining
independent variable. VIFs are determined by running a regression of each independent variable against the others, then the





VIF value is calculated as:

$$\mathrm{VIF} = \frac{1}{(1 - \mathrm{R}^2)},\qquad(2)$$

where $\mathrm{R}^2$ is the coefficient of determination (the proportion of the total sample variation in the dependent variable that is explained by the independent variables). Possible values of the VIF range from 1 to infinity. A VIF near 1 indicates there is no multicollinearity while VIFs between 1 and 5 suggest some multicolinearity but not enough to require adaption to the model. Values over 5 are cause for concern and values over 10 are a major problem and the model will require adaptation (Gareth
et al., 2013). Here the VIFs range from 1.01-1.62, well below the threshold of 5. We conclude that these independent variables do not suffer from significant multicollinearity and we can safely proceed.

### 3.1.2 Auto-correlation

Another assumption of linear regression is that the residuals (the difference between the regression model's predicted value and the actual value) are free from auto-correlation, i.e. that the residuals from the regression analysis are not correlated with
each other. Auto-correlation indicates that important information is missing from the model and the standard errors cannot be relied upon.

To test for auto-correlation, we use the Durbin Watson (DW) test, further details of the test can be found in Webster (2012). The output of the DW test is a number in the range [0, 4] (where square brackets indicate the closed interval). A result of 2 means that there is no auto-correlation present, results closer to 0 mean positive auto-correlation and towards 4 means negative
auto-correlation. Generally, values between 1.5-2.5 are expected, whilst results below 1 and above 3 can be cause for concern. As our linear regression analysis is composed of multiple models (a model for each height and month, for each wind component and for radar observations and WACCM-X model) we have multiple DW statistics.

For the radar data 82% lie in the interval [1.5, 2.5] and greater than 99% in [1, 3], for WACCM-X, 77% lie within [1.5, 2.5] and all DW statistics are within [1, 3]. Any deviation from a DW result of 2 indicates that some auto-correlation is present,
however, almost all of our DW statistics lie in the acceptable [1, 3] region. So whilst there will be some uncertainty over the standard errors for a minority ($< 1\%$) of models, the majority of models do not suffer from this problem.

### 3.1.3 Inclusion of a time term in the regression

Some regression studies such as Gan et al. (2017) include a time term in their regression equation to allow for a trend in time of the dependent variable, i.e. inclusion of a $\beta_6 \times t$ term in Equation 1. However, because Solar cycle 24 starts in 2008 at a
minimum and has its maximum in 2014, over the time period of our regression study (2005-2015) we have large positive correlation between the F10.7 index and time, namely multicollinearity between these two terms. When both terms were included in the regression analysis (not shown here), results suggested that the Solar term was artificially inflated and offset by the time term leading to unreliable results. The difficulty in separating time trends and the Solar cycle when the two are correlated is discussed in Qian et al. (2019).





To avoid this issue, we remove one of the correlated independent variables. Experiments with the regression analysis with a time term and no Solar term revealed lower $R^2$ values than the analysis with a Solar term but no time term. This indicates that the Solar term does more "explaining" than the time term and is the dominant influence over this time period. As a result, we chose to not include the time term and recognise that the results from the Solar coefficient may be slightly influenced by a long-term trend. Laštovička (2017) reviewed recent progress in trends in the upper atmosphere and noted that identifying

trends in atmospheric dynamics remains an open problem and that results vary depending on the location and time period. Beig (2011a) reviewed temperature trends in the mesopause region and found that new results indicated a negative trend whilst previous results had suggested no trend. Linear trends are notoriously difficult to separate from other long-term oscillations especially with observational data sets where long-term consistent measurements are hard to achieve and the trend itself may not be consistent.

## 4   Results: The winds in radar observations and WACCM-X

### 4.1   Zonal winds

The monthly-median zonal winds are presented in Figure 3, the first row shows the average year, with panel (a) the radar observations and (b) the WACCM-X simulation. In the observations, the zonal wind is characterised by the summertime wind reversal where the wind below heights of about 90 km becomes westwards throughout the Antarctic summer, maximising at

speeds of ∼-35  m s$^{-1}$ at the lowest heights observed. The eastward winds maximise in February, above the zero-wind line at heights of 95-100 km. The summertime wind reversal in WACCM-X has a similar temporal pattern, with the reversal to summertime westward winds beginning in October. However, the westwards winds in the WACCM-X reversal are stronger, maximising at -45 m s$^{-1}$ at heights near 80 km. Above the zero-wind line there are stronger eastward winds in WACCM-X than in the radar observations. Perhaps the most notable difference between the observations and the model occurs in April-

October at heights of 85-100 km; here WACCM-X predicts westward winds of up to 15 m s$^{-1}$, but the radar observations reveal eastward winds of magnitude 10 m s$^{-1}$. In summary, the three main differences in the zonal wind between the observations and the model are as follows:

- For wintertime winds (April - October) at heights of 85-100 km, the radar observations show eastward winds of magnitude 10 m s$^{-1}$, whilst WACCM-X has westward 15  m s$^{-1}$ winds.

- The maximum wind speeds in the summer wind reversal are weaker in radar observations than in WACCM-X.

- Above the zero-wind line of the summer wind reversal, WACCM-X winds reach speeds above 40 m s$^{-1}$ whereas in the radar observations they are weaker and maximise at only about 25 m s$^{-1}$.

The full timeseries of monthly-median winds for the radar and WACCM-X, respectively, are shown in panels (c) and (d). The seasonal summertime region of zonal wind reversal evident in WACCM-X winds is consistently both stronger and temporally

shorter than that observed by the radar. Further, we draw attention to the zonal wind wintertime bias that appears every single

**Figure 3.** Zonal monthly median winds as a function of height and time. (a) Radar observations average year, (b) WACCM-X predictions average year, (c) Monthly median winds from the radar observations for the interval 2005-2020, (d) Corresponding monthly median winds from WACCM-X for the interval 2005-2020. Solid black line indicates the zero-wind contour line and the contour interval is $10\,\mathrm{m\,s^{-1}}$.

year as a persistent feature, i.e., wintertime winds at heights of 85-100 km are observed by the radar to be eastwards but are predicted by WACCM-X to be westwards.





## 4.2 Meridional winds



**Figure 4.** Meridional monthly median winds as a function of height and time. (a) Radar observations average year, (b) WACCM-X predictions average year, (c) Monthly median winds from the radar observations for the interval 2005-2020, (d) Corresponding monthly median winds from WACCM-X for the interval 2005-2020. Solid black line indicates the zero-wind contour line and the contour interval is $10\,\mathrm{ms}^{-1}$.

In Figure 4 we show the meridional component of the wind, with (a) and (b) presenting the average year for the radar and
WACCM-X respectively. In the radar wind observations we see the upper branch of the Brewer-Dobson circulation at heights





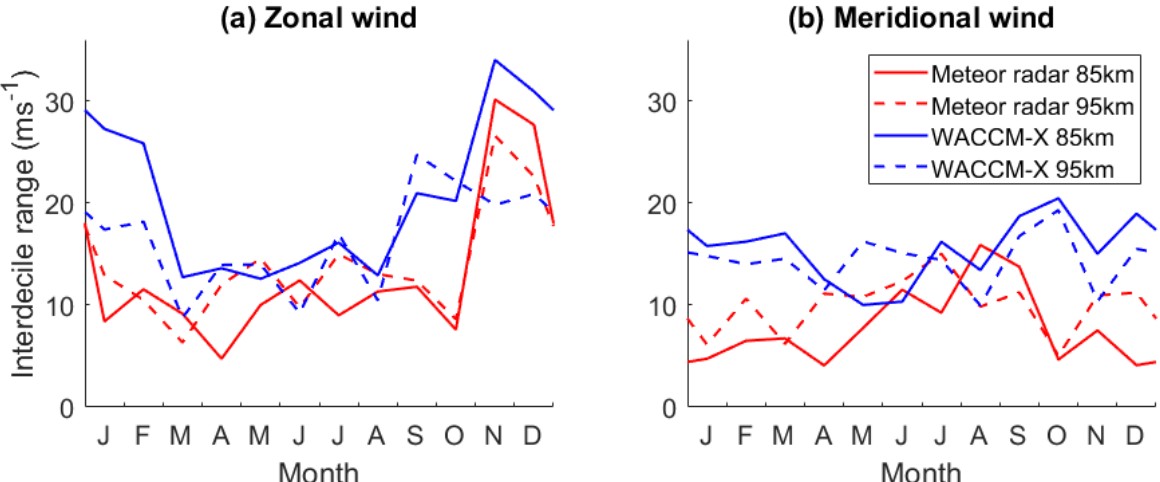

**Figure 5.** The interdecile range (90th percentile minus 10th percentile) of zonal and meridional winds for each month. Figures show radar data (red) and WACCM-X data (blue) at two heights, 85 km (solid line) and 95 km (dashed line).

of 80-90 km where in winter the wind becomes southwards. Throughout summer, the winds are northwards, flowing away from the pole. In the WACCM-X predictions, we also see some of the upper branch of the Brewer-Dobson circulation with generally southward winds during March-October although lower in height than seen in the radar observations.

As with the zonal component, we also note a persistent bias in the meridional winds when comparing the observations to the
model. However, this occurs in the summer (unlike the zonal wind bias which occurs in wintertime). In particular, in summer, at heights of 90-100 km WACCM-X predicts southwards flow, but the radar results reveal northward winds throughout the whole year at this height. In panels (c) and (d) we can see that this difference is present in all years.

To summarise, the two main differences between the radar observations and WACCM-X predictions in the meridional component are,

– During Antarctic summer at heights above 90 km the wind observed by the radar is northwards but in the WACCM-X simulation it is southwards.

 – The region of southward winds in winter reaches greater heights in the radar observations than predicted by WACCM-X.

**4.3 Interannual variability**

As can be seen in Figures 3 and 4, panels (c) and (d), there is significant interannual variability evident in both wind compo-
nents and in both the radar results and the WACCM-X predictions. In the zonal component, the shape and magnitude of the summertime wind reversal and the eastward winds above it change every year. The meridional winds are also variable and the height of the seasonal wind reversal changes every year, e.g., in 2015 it does not reach a height of 90 km but in 2005 it extends upwards beyond 100 km.



Figure 5 shows the interdecile range of each month at heights of 85 km and 95 km, for the radar and WACCM-X, and for

zonal and meridional components. The interdecile range is the 90th percentile minus the 10th percentile, which gives a measure of the variation in the monthly median wind speeds across the years considered. In (a), it can be seen that the interdecile range of the zonal wind maximises over the summer, when the zonal wind reversal occurs, due to the considerable variability in the strength and timing of this reversal. For both heights, there is more variability in WACCM-X than the radar observations. In (b), the interdecile range of the meridional winds observed by radar peaks in the winter, indicating variations in the strength of

the upper branch of the Brewer-Dobson circulation. In both components the interdecile range in the radar observations is less than that of WACCM-X.

## 5   Results: Linear regression analysis

### 5.1   The 11 year Solar cycle

In Figure 6 we present the results from the linear regression analysis for the zonal wind for the years 2005-2015. The first row

shows the results from the Solar coefficient i.e. $\beta_1$ from Equation 1. We scale $\beta_1$ by the interdecile range of F10.7 (defined as $\alpha$, here $\alpha = 70$ sfu), to give the wind response in units ms$^{-1}$ per $\alpha$. This quantity should be interpreted as the difference we see in wind speeds for a 70 sfu increase in F10.7. To allow for direct comparison of the magnitude of the influence of different external drivers we use the scaling by interdecile range as a standard measure of the spread of the indices. Hatched regions show where the relationship is statistically significant at the 90% level, using the Student's t-test. Each regression coefficient

value and statistical significance level is calculated using the regression method applied to data from a three-month window and presented on the figure at the location of the centre month.

In the regression results from the radar zonal-winds for the Solar term (Figure 6a), the biggest area of significance is in November and December at heights of 80-95 km, where we see a response of up to 9 ms$^{-1}$ per $\alpha$ eastwards, i.e. the linear regression fit to the winds suggests that were F10.7 to increase by 70 sfu the winds would be 9 ms$^{-1}$ more eastwards. This

positive correlation is present from October to January but is statistically significant during November and December. This result suggests that the increased irradiance during Solar maximum weakens the summertime westwards wind reversal and increases the strength of the eastwards winds above. This feature is also found in the linear regression results from the WACCM-X zonal winds, with a region of positive wind response in November and December, although it only reaches 90 km in height. This (like the radar results) weakens the strength of the westward wind reversal occurring at this time.

Another, smaller significant area with a negative response is found in March at heights of 82-96 km of magnitude 2-3 ms$^{-1}$ in the winds observed by the radar and at similar heights and magnitude in April in the WACCM-X predicted winds. This corresponds to the westwards winds near heights of 80-85 km persisting longer into March during times of higher F10.7.

Despite the similarities between the WACCM-X and the radar Solar coefficients in the zonal component, the meridional results are not alike (Figure 7a and 7b). In the radar observed winds we see a small statistically significant negative influence

at heights of 90-100 km throughout the summer and a mostly positive influence in the winter, although non significant. In the results from WACCM-X, we see a strong positive correlation between meridional winds and F10.7 beginning around the

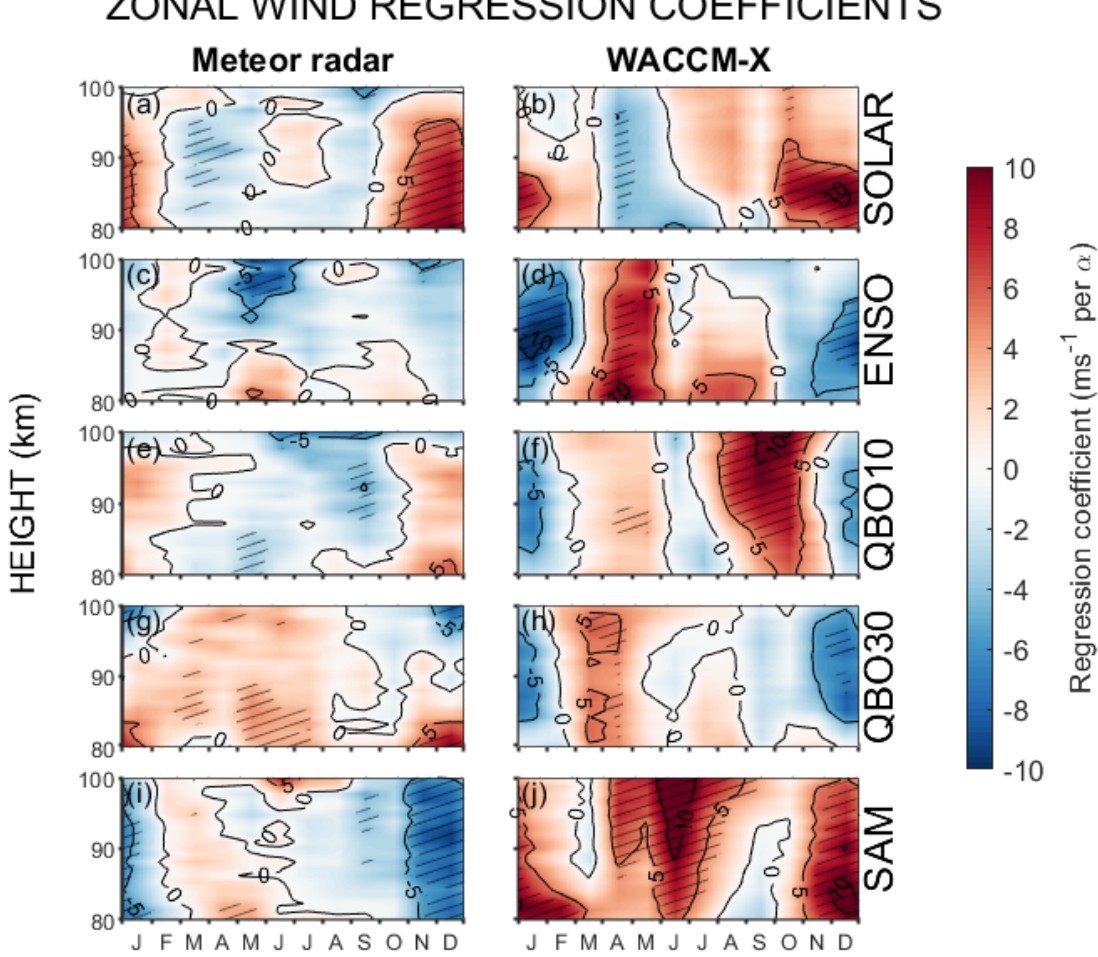

**Figure 6.** Regression coefficient results from the linear regression analysis (time period 2005-2015) for the zonal winds. Left column is results from radar observed winds, right column is from WACCM-X predicted winds. First row is the coefficient results from the Solar term in the linear regression, second row ENSO term, third row QBO term, fourth row QBO30 term and bottom row SAM term. Units are ms$^{-1}$ per $\alpha$. Where $\alpha$ takes a different value for each index as described in Table 1. Hatching covers regions statistically significant at the 90% level. Each regression coefficient and statistical significance level is determined using data from a three-month window, stepped by one month, and displayed at the location of the centre month.

spring equinox and lasting until March (although the response is only significant until December). This correlation is present at all heights. It reaches magnitudes of 6 ms$^{-1}$ per $\alpha$, i.e. a large influence given the typical wind speeds in the meridional component.

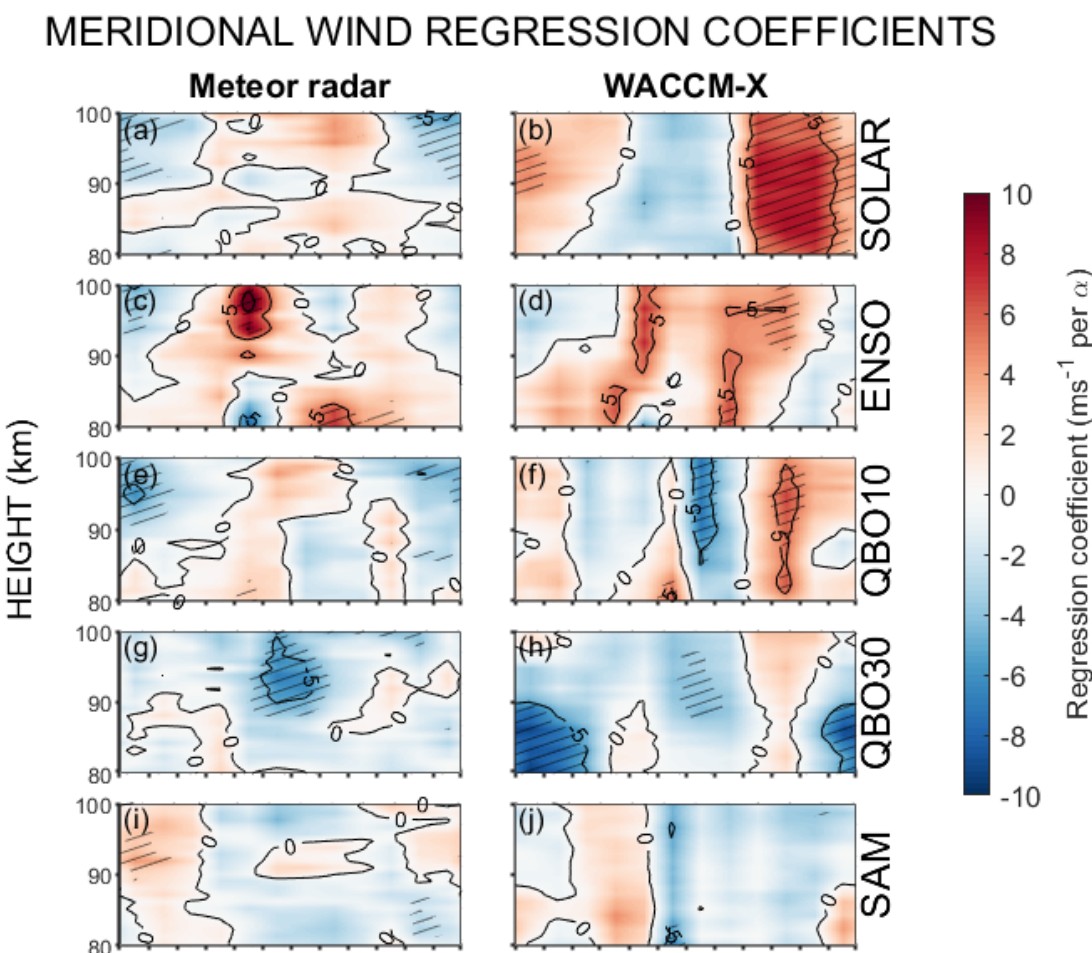

**Figure 7.** Regression coefficient results from the linear regression analysis (time period 2005-2015) for the meridional winds. Left column is results from radar observed winds, right column is from WACCM-X predicted winds. First row is the coefficient results from the Solar term in the linear regression, second row ENSO term, third row QBO term, fourth row QBO30 term and bottom row SAM term. Units are ms$^{-1}$ per $\alpha$. Where $\alpha$ takes a different value for each index as described in Table 1. Hatching covers regions statistically significant at the 90% level. Each regression coefficient and statistical significance level is determined using data from a three-month window, stepped by one month, and displayed at the location of the centre month.

## 5.2 The El Niño Southern Oscillation

In the results from the ENSO coefficient from radar observations (Figure 6c and 7c) at heights of 95-100 km and below 85 km, we see large values of this coefficient in both zonal and meridional components during April-May. This implies that there are stronger north westwards winds at the upper heights and a south eastwards winds around 80 km when the ENSO index is large




i.e. an El Niño event. Whilst this does not appear as a statistically significant response in WACCM-X in either component, we
see a smaller similar pattern in May in the meridional component. In fact, the meridional response to ENSO is fairly similar in
the radar results and WACCM-X results (see Figure 7c and 7d).

Despite the close agreement in the meridional, the zonal component exhibits no clear agreement. In WACCM-X in the zonal
wind a strong negative influence on the winds is seen from November through to February at heights of 85-95 km of magnitudes
$6\,\mathrm{ms}^{-1}$ per $\alpha$ and a positive influence of similar magnitude in April and May.

**5.3    The Quasi-Biennial Oscillation**

There is little agreement between the coefficient results from the radar and WACCM-X in either zonal or meridional compo-
nents for QBO10 (Figures 6e, 6f, 7e, 7f). In the zonal results from the radar observed winds (Figure 6e), when the QBO10
index is larger, the results suggests a generally eastwards influence on the winds in the summer and a westwards wind influence
in the winter, however, the significant regions are small. In WACCM-X (Figure 6f) we see a large significant positive wind
correlation in August-November, opposite to the negative correlation in the radar for September. The response in WACCM-X
over summer (December-February) is westwards i.e. opposite to the radar.

In the coefficient results for QBO30 there are some similarities between the radar wind results and WACCM-X results. In
the zonal component (Figure 6g and 6h) and when the QBO30 index is higher, we see wind that is generally more eastwards
in the winter and more westwards in the summer (at heights of 90-100 km). In the meridional component (Figure 7g and
7h) we also see agreement between the regression results from the radar observations and WACCM-X. Here, an increase in
the QBO30 index suggests more southwards winds with significant regions in June-August at heights of 90-100 km in both
observations and simulations. The WACCM-X meridional results also see a stronger, significant, southwards wind response
during December-March below heights of 90 km.

**5.4    The Southern Annular Mode**

The bottom row in Figures 6 and 7 shows the regression coefficient results from our final index, the Southern Annular Mode
(SAM). The SAM index exhibits its largest correlation with the winds in the zonal component (Figure 6i and 6j) whilst the
meridional results see very few significant areas (Figure 7i and 7j).

In the zonal winds, we find conflicting correlations between the observed winds and WACCM-X simulation. In the observa-
tions, we see the strongest response in November-January, where the results show that for increase of $\alpha = 2.5\,\mathrm{hPa}$ in the SAM
index, the zonal winds are $9\,\mathrm{ms}^{-1}$ more positive. This contributes to a stronger-reaching and higher-reaching summertime jet.
However, the linear regression results from WACCM-X predict the opposite response, where an increase of 2.5 hPa in the SAM
index correlates to a weakening of the jet by as much as $10\,\mathrm{ms}^{-1}$ and strengthening the eastwards winds above by $8\,\mathrm{ms}^{-1}$. Ad-
ditionally, WACCM-X zonal wind results find a large significant eastwards correlation throughout June and April-July above
90 km.

## ZONAL GRAVITY WAVE TENDENCIES FROM WACCM

**Figure 8.** Zonal mean zonal gravity wave tendencies from WACCM-X. Panel (a) shows the average year of zonal winds over 2005-2015. Panels (b)-(f) show the results of applying the linear regression method to the zonal mean zonal gravity wave tendencies with (b) the Solar coefficient, (c) ENSO, (d) QBO10, (e) QBO30 and (f) SAM. Hatching covers regions statistically significant at 90%. Each regression coefficient and statistical significance level in panels (b)-(f) is determined using data from a three-month window, stepped by one month, and displayed at the location of the centre month.

## 6 Gravity Wave Tendencies

Figure 8 shows the zonal mean zonal gravity wave tendencies from WACCM-X. In panel (a) we present the average year. Throughout Antarctic winter there are negative gravity wave tendencies and in summer stronger positive gravity wave tendencies. The strength of the gravity wave tendencies are stronger at the lower heights, maximising at $100\,\mathrm{ms}^{-1}\,\mathrm{day}^{-1}$. Coefficient results from regression (same method as used for the winds) are shown in panels (b)-(f). For the Solar cycle term (Figure 8b) in November to January at heights of 80-90 km, we see a decrease of $\sim 10\,\mathrm{ms}^{-1}$ per day for an increase in 70 sfu. These significant regions line up with timing and heights of significant and stronger eastwards winds seen in the linear regressions results for the Solar term (Figure 6b).

In Figure 8, panels (c),(e) and (f) we see the regression results from the ENSO, QBO10 and SAM respectively. All three of these results show minimal significance regions across the year and at all heights. Figure 8d presents the QBO10 coefficient, which unlike the others, has almost half the year (July - December) seeing a significant change in the zonal mean zonal gravity wave tendencies in the WACCM-X model. Again this negative correlation in the gravity wave tendencies corresponds to a large eastwards correlation in the zonal winds (Figure 6f) during times when the QBO10 index is large.



## 7 Discussion

### 7.1 Wind observations

In our study we have presented the first long-term radar wind observations from Antarctica made by a meteor radar with height resolving capabilities. We can make specific comparisons to other studies. Dowdy et al. (2007) presented climatologies of observations made by MF radars at Davis (69°S, 78°E, 1994-2005) and Syowa (69°S, 40°E, 1999-2003). Their zonal wind observations revealed some general similarities in the seasonal variability with our meteor radar results. However, there are notable differences. The MF radar observations of zonal winds show weaker eastwards winds above the zero wind line (5-

10 ms$^{-1}$ in the MF radar winds compared to 10-20 ms$^{-1}$ in the meteor radar winds in January at heights of 95-100 km).

Baumgaertner et al. (2005) presented wind observations from 1985-2004 made by an MF radar (heights of 75-95 km) at Scott Base (78°S, 167°E). In their study they found that there is a large degree of interannual variability in the wind speeds, noting that the interannual variability peaks in December zonal winds, in good general agreement with our results. We find that the climatologies presented by Baumgaertner et al. (2005) had much lower wind speeds in both zonal and meridional components,

with the zonal summertime wind reversal reaching a maximum of 18 ms$^{-1}$ in their results compared with ~30 ms$^{-1}$ in our results. Additionally, although the data only extends to 95 km in height, the zero wind line in the zonal winds is higher in Baumgaertner et al. (2005) (95 km in Janurary compared to 90 km in the meteor radar results) and the eastwards winds above very weak. This is a similar difference to that seen when we compared our meteor radar to Dowdy et al. (2007).

Another study of interest is Hibbins et al. (2005), where wind observations for 1997-1998 and 2002-2004 from the MF radar

at Rothera (i.e. the same location as our meteor radar) were presented. The average year presented by Hibbins et al. (2005) shows some general agreement with our average year from the meteor radar observed winds. However, the zero wind line in both the zonal and meridional winds is far lower in height in Hibbins et al. (2005) than our results in this work. For example, in January the zonal zero wind line is 6 km lower in Hibbins et al. (2005) and in June the meridional zero wind line is 8 km lower. As noted by Sandford et al. (2010) this is significantly different to both Antarctic MF and meteor radar climatologies and that

later analysis of the Rothera MF-radar data from 2005 onwards shows a much higher summertime zero-wind line, closer to the results of other MF radar studies. We suggest that this may indicate a bias in height estimates in data from the Rothera MF radar recorded before 2005.

In summary, we find that there are differences between our meteor radar observed winds and the MF radar winds from Baumgaertner et al. (2005), Dowdy et al. (2007) and Hibbins et al. (2009). These discrepancies may arise due to interannual

variability, longitudinal differences or biases between the different techniques. Here, we draw attention to Manson et al. (2004) and Jacobi et al. (2009) where the differences between MF radar and meteor radar observed winds for collocated instruments were quantified. In general, although there is good agreement at heights of 80-90 km, MF radars were found to record significantly weaker winds at heights above 85-90 km than were recorded by meteor radars. Wilhelm et al. (2017) also explored the differences between the meteor radar and MF radar winds. They attribute the differences to two potential reasons, both arising

from the MF radar technique. Firstly, differences between the measured centre of scatters and the true centre in the MF radar





beam and secondly, side lobe contamination. This highlights the suitability of MF radar studies for winds below 80 km and meteor radar for above 80 km.

Next we consider comparisons between our radar observations and the WACCM-X simulations also presented in this work. We find that although there is agreement in some periods of the seasonal cycle, WACCM-X has persistent biases that occur ev-

ery year. Specifically, in Antarctic winter at heights of 85-100 km, our radar observations reveal eastwards winds (of magnitude $\sim$10 ms$^{-1}$) and yet WACCM-X predicts westwards winds (of magnitude $\sim$15 ms$^{-1}$) i.e. winds in the opposite direction. The observed and WACCM-X meridional wind results also differ in the summer at heights of 90-100 km. The radar observations reveal northwards winds ($\sim$5-10 ms$^{-1}$) yet WACCM-X has southwards winds ($\sim$5 ms$^{-1}$). We propose that the persistent bias in zonal wintertime winds in WACCM-X is due to the lack of secondary gravity wave modelling in WACCM. Dempsey et al.

(2021) presented radar results from 2009 from Rothera and showed that both WACCM and eCMAM (the extended Canadian Middle Atmosphere Model) suffers from the same, westward (negative) winds in the winter for this year. Our work builds on this and shows that the discrepancies are persistent biases occurring every year in our dataset. Becker and Vadas (2018) propose that the missing eastwards momentum in the models could be generated by secondary gravity wave breaking in the atmosphere; consistent with this hypothesis, the HIAMCM (the HIgh Altitude Mechanistic general Circulation Model) they

developed includes secondary gravity waves and correctly predicts the mesospheric wintertime eastward flow seen in the radar zonal climatology.

## 7.2 Linear regression results

### 7.2.1 The 11 year Solar cycle

Beig (2011b) summarised studies of Solar cycle influence on MLT temperatures, and found that a Solar maximum increases

temperatures by a few K per 100 Solar flux units (sfu), and that this change increased to about (4-5 K per 100 sfu) at the upper heights of the MLT region. Later, a combined modelling and observational paper Gan et al. (2017) found significant temperature responses in SABER (Sounding of the Atmosphere using Broadband Emission Radiometry), a satellite-based instrument and eCMAM (extended Canadian Middle Atmosphere Model) using a linear regression method. They found a significant correlation between Solar maximum and a hotter atmosphere in the middle and upper atmosphere, with the temperature response

increasing with height. This is in agreement with the temperature responses summarised by Beig (2011b) and found by Ramesh et al. (2020) and Cullens et al. (2016) using the WACCM model. The wind responses in Gan et al. (2017), Ramesh et al. (2020) and Cullens et al. (2016) are less statistically significant and more variable than the temperature response. This is perhaps due to the complex wave processes governing the winds in the MLT. However, all three studies find a significant relationship between the Solar cycle and the Antarctic MLT winds, at least during some seasons. This is in general agreement with our

results.

In our results we find the largest positive correlation between the zonal wind and F10.7 at heights of 80-95 km in spring and early summer, this correlation is found in both our radar observations and the WACCM-X simulations (localised at Rothera). We compare this to Ramesh et al. (2020) where WACCM6 temperature and wind results were regressed against indices for





F10.7, QBO, ENSO, EESC (Equivalent Effective Stratospheric Chlorine), $CO_2$ and AOD (area-weighted global average of stratospheric aerosol optical depth at 550 nm). We find that, for the latitude of Rothera, our result of a positive correlation of F10.7 with the zonal winds agrees with Ramesh et al. (2020). Although the response in Ramesh et al. (2020) is $\sim1$ ms$^{-1}$ per 100 sfu, much smaller than the response of $\sim$5-10 ms$^{-1}$ per 70 sfu found in our results. A possible reason for this magnitude difference could be the time period used: Ramesh et al. (2020) used a much longer interval (1850-2014) whilst we are restricted by available observational data. Additionally, Ramesh et al. (2020) considered zonal-mean zonal-winds, whilst our radar and analysis of WACCM-X data are localised to the zonal winds at Rothera. Smith (2012) states that due to planetary scale oscillations in monthly averaged winds, the local time-averaged zonal wind is likely to differ from the zonal-mean zonal-wind.

Cullens et al. (2016) also used WACCM to explore the influence of the Solar cycle on temperature and wind in the atmosphere. Cullens et al. (2016), like Ramesh et al. (2020) used zonal-mean zonal-wind values. For the latitude of Rothera, our results disagree with those from Cullens et al. (2016) who found that the winds are significantly more westwards during a Solar maximum than a Solar minimum during September-October and with no significant change in December-January. We propose that our WACCM-X results differ to that of Cullens et al. (2016) due to the different time periods analysed. Cullens et al. (2016) used simulated data for 1955-2005; not only is this a far longer period, incorporating 5 Solar maxima, it has no overlap with our study. We note that for our work in this study, in the zonal component, the WACCM-X data and radar observations taken over an identical time period and location, mostly agree with each other.

Baumgaertner et al. (2005) examined observational wind data from 1985-2004 from an MF radar at Scott Base, Antarctica. They found no significant correlation between the Solar cycle and winds (when partitioning for QBO phase); however, this station is significantly further poleward than Rothera. Wilhelm et al. (2019) fitted an 11-year period sine wave (unconstrained in phase) to radar winds from three northern hemisphere sites in the Arctic. Andenes at (69°N, 16°E) is at conjugate latitudes to Rothera and Wilhelm et al. (2019) found the strongest amplitude response in summer (2-4 ms$^{-1}$ at heights of 80-85 km in June-August). This is in general agreement with our results and suggests that similar correlations with the Solar cycle may be evident in both the Arctic and Antarctic, although differences in methodology preclude further comparisons in the sign of this response. In the meridional component Wilhelm et al. (2019) found little significant change in the winds in disagreement with our results.

Whilst the zonal wind results from the linear regression are similar between the radar observations and the WACCM-X simulations within this study, the meridional wind results are very different. This suggests that the Solar influence on the upper branch of the Brewer-Dobson circulation may not be properly simulated in WACCM-X. This could be due to the lack of secondary waves in WACCM-X or the altitude bias in the modelling meridional winds evident in figure 4.

Finally we consider the gravity wave tendencies from the WACCM-X run. We see a significant negative response of around 10 ms$^{-1}$ per day in zonal gravity wave tendency during November to early February at heights of 80-90 km. This is notable as it lines up with the times of positive wind response to the Solar cycle in the zonal wind in both the radar and WACCM-X results. We recall that the WACCM-X winds are localised to Rothera, whilst the gravity wave tendencies are a zonal mean around the a latitude belt centred on Rothera. Therefore results do not strictly correspond to the similar wind results. During





winter, Rothera lies under a region of peak gravity wave activity that results from winds blowing over the Antarctic Peninsula

so the zonal mean may be more representative of local gravity wave tendencies, but this is not the case for summer.

### 7.2.2  The El Niño Southern Oscillation

In the radar observations, the regression results from the ENSO coefficient have far fewer significant areas than the Solar results, leading us to conclude that there is little linear influence on the MLT winds by ENSO. Li et al. (2016) explore the response of the summer southern hemisphere to ENSO, proposing a mechanism wherein Northern Hemisphere planetary wave activity is

increased during an El Niño event causing a chain of changes leading to anomalous southern hemisphere mesospheric eastward gravity wave forcing. Llamedo et al. (2019) also found a 3.5-4 year oscillation in lidar observed gravity wave activity in the stratosphere in Tierra del Fuego that they suggest could be an ENSO influence. Ramesh et al. (2020) found that the zonal mean winds have the strongest and geographically largest regions of correlation to ENSO at mid latitudes. Whereas at higher latitudes, the correlation is weaker but still significant at times. We see in our radar results the strongest response in May,

however, Ramesh et al. (2020) does not present results for May and so we are unable to compare.

In contrast to the wind observations from the radar, the ENSO coefficient in our analysis of the WACCM-X zonal winds is large and significant during autumn and summer, however, the meridional results see some similarity between the observations and WACCM-X.

For the gravity wave tendencies from WACCM-X, there are no significant results suggesting that in the model gravity waves

do not play a role in the linear ENSO influence on the winds. It is expected, however, that gravity wave interactions that link ENSO to the Antarctic MLT winds are likely to involve non linear processes and hence would not be observable in the results in this study.

### 7.2.3  The Quasi-Biennial Oscillation

For the QBO results there is little agreement between the radar results and the WACCM-X observations. The wind responses

to the QBO10 and QBO30 index also show no notable similarities with each other, however, the indices were initially chosen to be orthogonal, so this is expected. Ramesh et al. (2020) also used a QBO10 and QBO30 term in their linear regression analysis of WACCM-X zonal mean zonal winds. Their results showed the QBO response as a primarily equatorial response that did not extend above the stratopause throughout June-August. In their MLT results, in December-February the QBO has an influence on the equatorial region and the northern hemisphere, but not the southern hemisphere. Ford et al. (2009) explored the

QBO effects on Antarctic mesospheric winds using an Imaging Doppler Interferometer at Halley (75.6°S, 25.5°W) and found that different lags of the QBO at 50 hPa correlated with wind affects at different times of the year in the mesosphere. They proposed that the winds are modulated by action of the equatorial QBO on planetary wave activity. We see little agreement when comparing the results with Ford et al. (2009) at similar QBO pressure levels; at 7 hPa to our 10 hPa (QBO10) and the 23 hPa to 30 hPa (QBO30). However, this may be a product of the different time periods used, different location or different

techniques. In contrast, using MF radar winds from Scott Base, Antarctica, Baumgaertner et al. (2005) found no consistent relationship between MLT winds and the QBO using a variety of techniques.





In the gravity wave tendency results from WACCM-X we see a significant negative correlation between the zonal mean zonal gravity wave tendencies and the QBO10 for July-December at all heights, this suggests that there may be modulation of gravity waves with the QBO in WACCM-X.

Like Ford et al. (2009), our regression results show that there is a significant influence from the QBO on the Antarctic MLT at times. However, due to the nature of the QBO with its vertical structure of descending phase fronts, the full influence of the QBO on MLT winds cannot be characterised by only considering the two indices. In this study, the inclusion of the QBO10 and QBO30 indices serve to identify that there is a change in the winds during times of opposing QBO phases and to remove the main effects of the QBO from the other results. More research is needed to understand the processes behind this link between the QBO winds and the Antarctic MLT dynamics.

### 7.2.4 The Southern Annular Mode

There have been very few previous studies on the influence of the SAM on Antarctic MLT winds. Merzlyakov et al. (2009) found no significant correlations between the winter zonal wind and the SAM index with a composite dataset of radar observations (with no height resolving capability) for 1970 to 2006 across a variety of Antarctic sites. This is consistent with our radar observations where there is little significant influence of the SAM on the zonal winds in winter. However, in winter WACCM-X results there is a strong, significant correlation between the zonal winds and the SAM index that is not present in observations. This may be due to the strength of the stratospheric polar vortex being around $10 \, \text{ms}^{-1}$ too strong in WACCM-X (Butchart et al., 2011), changing the feed-backs between the SAM and the polar vortex and changing the surface-mesosphere coupling pathways.

In summer, from observations, we see a large negative correlation between the zonal wind and the SAM index. In WACCM-X we also have a large correlation but it is positive at this time. The sign of the zonal wind results from the observations and WACCM-X directly contradict each other. In the meridional component there is little significant correlation suggesting that wind variations with the SAM are primarily zonal.

### 8 Conclusions

In this work we have carried out the first long-term study of the interannual variability of Antarctic MLT winds using the meteor radar at Rothera on the Antarctic Peninsula. We have presented a multi-year climatology of meridional and zonal winds and compared these meteor radar observations to predictions from WACCM-X. Further, we have explored the interannual variability of both observed and modelled winds and gravity wave tendencies with a multilinear regression model, regressing against five key indices, namely; the 11-year Solar cycle, El Niño Southern Oscillation (ENSO), two indices for the Quasi-Biennial Oscillation (QBO) and the Southern Annular Mode (SAM). We find that there are notable differences between the observed and modelled winds in both the multi-year seasonal climatology and in their interannual variation with climatological indices.

From our work, we conclude that:





1. Persistent biases are found between observations and the WACCM-X model. In particular, the observations reveal east-ward winds of ∼10-15  m s$^{-1}$ in wintertime (April-September) at heights from about 85-100 km, but at these heights WACCM-X predicts westward winds of ∼15 m s$^{-1}$. In the meridional component, summertime (October-March) winds at heights from 90-100 km are observed to be northwards but are predicted to be southwards in WACCM-X. We propose that the biases in wintertime zonal winds are due to the lack of eastward forcing from secondary gravity waves in WACCM-X.

2. Both the observed and model winds show a significant degree of interannual variability with the highest interdecile range in the summer (October-March) zonal winds, due to variations in the strength and height of the summertime wind reversal.

3. In summer (October-March), the observed zonal winds vary with the Solar cycle. In particular, at heights of 80-95 km these winds are up to 9 m s$^{-1}$ more eastwards for an increase in Solar activity of 70 sfu. This agrees with the WACCM-X predictions for heights of 80-90 km. During summer at these heights, the zonal gravity wave tendencies in WACCM-X have a negative correlation with F10.7.

4. The variation of observed wind with the Niño3.4 ENSO index is sporadic and generally less significant than for the other indices, suggesting that there is no linear ENSO influence on the MLT Antarctic winds.

5. Determining the wind response to the QBO is challenging with a linear regression method due to the characteristic descending phases of eastwards and westwards winds. In our results, we find differing responses to the QBO between the observations and the model and depending on the height used to capture the QBO behaviour.

6. The relationship between MLT winds and the SAM has not previously been explored in detail. Our results suggest that both the observed and model MLT zonal winds do vary strongly and significantly with SAM in the summertime (October-March). However, the sign of this response is opposite between the observations and WACCM-X.

7. Further, the WACCM-X wintertime zonal winds are significantly more positive when the SAM index is high, but the observed zonal winds see no significant change in winter. We propose that this difference could be a result of the biases in the model's stratospheric polar vortex altering the critical level filtering of gravity waves from below.

This work highlights the importance of using observations to constrain GCMs as they are extended upwards into the mesosphere and above.

*Code and data availability.* The meteor radar data used in this work is from Mitchell, N. (2019): University of Bath: Rothera Skiymet Meteor Radar data (2005-present). Centre for Environmental Data Analysis, 2020. https://catalogue.ceda.ac.uk/uuid/aa44e02718fd4ba49cefe36d884c6e50. The WACCM-X data is from NCAR Climate Data Gateway SD/WACCM-X V. 2.1 EXTENDED RUNS (1980-2017) Gasperini (2019b). Data for F10.7 is provided by the National Research Council of Canada http://www.spaceweather.ca/solarflux/sx-4-eng.php. Figure code and data is available at P E Noble (2022)



535 *Author contributions.* The study was proposed and designed by TMG, NM, CW, SE, CC, PN. Data analysis was led by PN with radar hourly winds supplied by NH. Model data was processed by CC. Manuscript and all figures prepared by PN. Scientific interpretation led by PN and contributed to by all authors

*Competing interests.* The authors declare that they have no conflict of interest

*Acknowledgements.* PN is supported by a NERC GW4+ Doctoral Training Partnership studentship from the Natural Environment Research
540 Council (NE/S007504/1). CW, NH, NM and TMG are supported by the UK Natural Environment Research Council (NERC) under grant numbers NE/R001391/1, NE/S00985X/1 and NE/R001235/1. CW is also supported by a Royal Society University Research Fellowship UF160545. CYC is supported by NSF 1855476. This material is based upon work supported by the National Center for Atmospheric Research, which is a major facility sponsored by the U.S. National Science Foundation under Co- operative Agreement 1852977. N. P. acknowledges support from NASA Grant 80NSSC20K0628.



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
