# Peer review of "Interannual variability of winds in the Antarctic mesosphere and lower thermosphere over Rothera (67°S, 68°W) in radar observations and WACCM-X"

_Atmospheric Chemistry and Physics, 2022_

## Community Comment (CC1)

**Review of the paper entitled "Interannual variability of winds in the Antarctic mesosphere and lower thermosphere over Rothera (67°S, 68°W) in radar observations and WACCM-X" by Noble et al.**

The authors present the long-term interannual variability of zonal and meridional winds in the mesosphere and lower thermosphere (MLT) mainly from meteor wind radar observations during 2005-2020 over an Antarctic station, Rothera (67°S, 68°W). They also compare the observed MLT winds with the NCAR's extended version of the Whole Atmosphere Community Climate Model, WACCM-X simulations during 2005-2017. Further, the monthly/seasonal variations in MLT horizontal winds in response to solar forcing, Quasi-Biennial Oscillation (QBO 10 hPa and 30 hPa), El Niño Southern Oscillation (ENSO) and Southern Annual Mode (SAM) are compared from radar observations with the model simulations during 2005-2015. Significant biases were found between observed and model climatological monthly winds. The authors propose these biases are due to the lack of gravity wave forcing in the model simulations. I have major concerns in the present form of this work. The authors did not explain any strong reason why they considered only the above mentioned forcings in their regression analysis? As the winds are related to temperature gradients and they are in close agreement with the thermal wind balance, the climate index for temperature variability or its driving factor like $CO_2$ or $O_3$ has not been considered. The gravity wave tendencies are discussed but not the tides as they are also the key drivers of the horizontal winds in the MLT region. The wind anomalies (mean removed) are used in the regression analysis, however it is not clearly stated whether the climate indices are used as their anomalies (mean removed). If not done so, the mean removed anomalies of the predictors must be used in the regression analysis. Please avoid explaining/focusing statistically insignificant regions of the results obtained from linear regression analysis.

Below are my specific comments (major & minor).

Line #4: Replace 'test' with other wordings like 'validate' or 'evaluate'.

Line #7: Here and anywhere else in the paper, the authors used the monthly median winds. How the monthly medians vary from monthly mean in the regression analysis? It will be more comfortable if elaborated the difference between them.

Lines #8-9: For which year (s) you are referring to?

Line #11: why only the secondary gravity waves?

Lines #17-18: Here and wherever applicable, what is meant by 'correlations' between predictand and predictors? Do you mean 'responses'?

Line #29-30: Include 'after they break' between 'their momentum' and 'and thus driving..'

Line #61 and 69: Do these both studies are represent 'first' results of this radar? Both can't be the 'first'.

Section 2.1: Include more technical details of the SKiYMET radar.

Line #98: Why only 20 meteors considered here as a minimum threshold? How this will impact the accuracy of wind retrievals…should explain.

Line #112: I didn't find the model winds shown for the period of 1980-2017. Is it for 2005-2017…correct it.

Section 2.3: Definitions of all the climate indices must be included.

Panel (b) of Figure 2: It seems the ENSO values (on y-axis) are in $^{\circ}$C and not in K….correct it. Perform the regression analysis again with the corrected values.

Section 3.1:

(i)     How many runs/realizations of the model have been used in this study? More runs/realizations enhance the statistical reliability.

(ii)    It is very important to use the anomalies of the climate indices in the regression model. If not done so, the anomalies must be used to repeat the regression analysis.

(iii)   In equation (1), what are $\beta_1$, $\beta_2$…..$\beta_5$ represent? Also U' stand for zonal or meridional wind anomalies? In general, U is the symbol to represent the zonal wind. As mentioned above, define all the climate indices used in the regression analysis.

Line #156: What are those 'other causes'….please include?

Section 3.1.1:

(i)     I am not sure why the VIFs are needed and calculated? The correlation coefficients (R) between the predictors must be included. As I see from Figure 2, no two predictors are correlated….verify it with the R values by including them.

(ii)    Line #172-175: The references for the values must be included.

(iii)   What are the VIF values for individual index? This helps for complete diagnosis.

Line #182: Include more details of DW test instead simply adding a reference.

Line #183-185: It is very important to include the reference for the values mentioned here.

Section 3.1.3:

(i)     What is the purpose of this section as the 'time' or 'trend' term is not included in the regression equation as there is no $\beta_6$ x t in equation (1). However the

multicollinearity between $F_{10.7}$ and time should be evidenced by a figure or values. Otherwise, it is difficult to understand for the readers.

(ii)     Line #200-201: What about the VIF values of other indices in the regression equation in this experiment? i.e. whether they are increasing or decreasing when one of the two correlated solar or time terms removed.

Line #211: Here and wherever applicable, does the 'average year' represent 'composite mean' of all the years? It should be replaced.

After line#228: Also the stronger winds occur in different months, in February from radar and in Dec-Jan in WACCM-X.

Section 4.1, Figure 3: Panels (a) & (b) – the radar zonal winds are averaged for 2005-2020 with gaps during 2016-2018 and, parts for 2009-2010. The model zonal winds are averaged for 2005-2020 with data gaps during 2018-2020. How accurate the composite zonal wind comparisons between radar observations and model simulations with these data gaps? This is misleading completely in making the conclusions of biases between observed and modelled winds. Moreover the regression analysis has been performed for the winds during 2005-2015 only. Please use the datasets for the same duration to avoid misperceptions.

Line #231-232: Replace 'to be' with 'are' & 'predicted' with 'simulated' (wherever applicable in the paper). Remove 'are' after 'eastwards but'.

Section 4.2, Figure 4: Refer to the above comment for Figure 3. The composite monthly means with the data gaps impose inaccuracies in drawing the important conclusions particularly comparing two datasets of observations and model.

Line #235, 237 & 260: Here and wherever applicable, are the 'upper branch of the Brewer-Dobson circulation' and the 'mesospheric meridional circulation' same or related to each other? I am confused with this terminology and their different driving mechanisms: former is accompanied by the planetary wave (equator-to-pole) and the latter is due to gravity wave driven circulation (pole-to-pole).

Line #250: Replace 'beyond' with 'up to'. Does this line represent 'model' or 'observation'?

Line #257: Replace 'over' with 'in'. The wind reversal is evident in WACCM only but not in the radar observations. But still why the interdecile range is maximum (above 90 km) in summer from radar observations also (Figure 5a)?

Line #259: The interdecile range of meridional winds doesn't peak in the winter from WACCM-X simulations…..why?

Section 5.1: Why the authors considered the solar response per 70 sfu? The normal practice to retrieve the solar response is per 100 sfu.

Line 269: Here and wherever applicable, why the statistical significance is not considered for 95% or 99% level? This is maybe more helpful than the 90% level.

Line #270: Why do you need to use three-month window instead monthly means?

Line #276: Do you mean the 'Solar maximum' refers to 'solar cycle maximum' or 'solar irradiance maximum'? What could be the physics behind why the solar maximum weakens the summertime westward wind reversal and strengthens the eastward winds?

Line #286: Do you mean 'positive response' at 'positive correlation'?

Sections 5.2, 5.3: Avoid explaining/focusing the regions which are not statistically significant.

Line #293 & 303: what is meant by 'index is large'? Does this have any significance?

Line #294-295: I could see significant positive and negative responses in model zonal winds in April and Dec-Jan respectively.

Line #301: I could see no agreement from radar and model.

Line #306: Replace 'over' with 'in'.

Lines #304-306: From figures 6e & 6f, I am wondering why the radar and model exhibit the opposite responses to QBO10 particularly in summer. Because both show the same westward winds below 90 km during this season (figure 3a & 3b) and thus we can expect both the responses should be same.

Line #308 & 337: In general, we see eastward and westward wind regimes in the QBO. But what does it meant by 'index is higher' or 'large' here?

Line #312-313: From figures 7g & 7h, why the meridional wind response to QBO30 is negative during Dec-Mar only in the model and not in the radar observations? Because the winds are northward from figure 4a & 4b, we can expect the similar responses in both radar observations and model simulations during this season.

Line #320: Is it more 'negative'? Please check. Does it mean more 'westwards'?

Section 6, figure 8: I am confused how the gravity wave tendencies are estimated? Also how these tendencies could explain the wind responses in WACCM-X (not sure why they are not shown for rocket observations).

Line #326: Not sure why the 'zonal mean' and not localised? This could lead to de risory comparisons. 'In panel (a) we present the average year' of what?

Line #333: Panel (e) represents QBO30 and not QBO10.

Section 7.1: I am not sure why the winds from two different techniques (meteor and MF radar) and during different observational periods are compared here. Also how these comparisons are scientifically useful as the changes in several dynamical/chemical processes in the atmosphere over time could lead to different results. Obviously it is not expected the winds measured during 2005-2015 (in this study) are comparable to those observed during 1994-2005, 1999-2003 and 1985-2004.

Line #371: Not sure whether the MF radar winds go below 80 km (or 78 km)? Any references?

Line #459: Remove 'observations'.

Line #462: It is WACCM6 and not WACCM-X.

Line #468: Replace 'similar' with 'nearly similar'.

---

## Author Comment (AC1)

**Response to Reviewers for "Interannual variability of winds in the Antarctic mesosphere and lower thermosphere over Rothera (67°S, 68°W) in radar observations and WACCM-X" by Noble et al**

We thank the reviewers and community comment for their detailed comments on our paper which have led to a much improved manuscript. We respond to the comments individually below.

**1 Reviewer 1**

**1.1 Main Comments**

1. Some aspects of the analysis method are novel but their rationale are not fully described. The authors often use the median to describe the data. At line 102, they say this removes tides and planetary waves but do not say why this is so. Such wave phenomena can lead to velocity distributions that have strange shapes. The authors' use of the median and the interdecile range to describe the data gets around this. In their comparisons with other results, though, they need to make sure the reader is cognizant of the use of different statistical measures.

Following this comment and a similar comment about the use of a monthly median from the community review, we have changed the analysis throughout the paper to monthly means instead of monthly medians. This is to allow for better comparisons with existing work. The changes to the results are minimal. Note - as well as the plots changing slightly, the DW statistics have changed slightly but the VIFs remain the same (as they are calculated on the independent variables which have not changed).

We have also added the following sentences to the text to justify the use of monthly averages in removing tides and planetary waves, along with clarification on the limitations of the monthly means. "From the composite days we calculate a monthly mean wind value in the zonal and meridional components. Taking the monthly mean removes these tides as tides have periods of 24, 12, 8 and 6 hours. Planetary waves have periods shorter than one month, so averaging over the month mostly removes them. Our analysis will not remove stationary planetary waves and while we acknowledge this is a possibility, we do not expect this to significantly affect our results."

For the interdecile range, we have added a description of this statistical measure in the text where it is first introduced in Section 2.3, so that the reader is familiar before it used in more detail in the results section.

2. In the abstract and the conclusions, the authors propose that a bias exists in WACCM zonal winds due to missing eastward momentum forcing. This conclusion can be supported on the time scale of a single time step (because on that scale, the model is using the forcing to define a grid point's next velocity value). But on longer time scales, the dynamical equations include the influence of other forcings such as the Coriolis effect. If balanced flow was assumed, the zonal wind bias would be due to incorrect meridional forcing. The authors need to consider their proposal here more carefully.

The reviewer raises an important point, and we apologise that our description was incomplete in this regard. Both WACCM and WACCM-X are known to exhibit a zonal wind bias in the winter mesosphere at high latitudes compared to observations [Dempsey et al., 2021, Hindley et al., 2022], which is what we also show in this paper.

Specifically, the model exhibits wintertime zonal winds that return to westward above around 80km altitude, while observations show they continue to be eastward throughout the MLT. Recently, this phenomenon has been explored at length, and it is thought that it is highly likely that missing eastward gravity wave momentum present in the atmosphere not included in the model is responsible for the bias [Vadas and Becker, 2018, Becker and Vadas, 2018, Vadas et al., 2018, Heale et al., 2020, 2022].

However, as the reviewer suggests, it is not so simple as a single eastward momentum term being added to the circulation at one point to drive the flow. The proposed process has several steps, as discussed by Becker [2012]. The general proposed process (which occurs in the real atmosphere but is missing in WACCM-X)

is: (i) Secondary gravity wave breaking in the MLT drives a residual circulation that leads to (ii) upwelling over the polar cap (and, for completeness, mass-conserving downwelling at lower latitudes), which leads to (iii) a meridional temperature gradient, which leads to (iv) an eastward vertical wind shear, which in turn is visible as (v) an eastward wind in the winter MLT at high latitudes. Without the first step of the proposed secondary gravity wave breaking in the MLT in WACCM (which has only primary westward GW forcing in the MLT due to wind filtering below), this proposed process does not occur and the model winds return to westward above 80km during winter.

This is the process that we propose may lead to the wintertime zonal wind discrepancies we observe between the radar and WACCM-X, and we have added this information to the manuscript. However the process is complex and further exploration is perhaps beyond the scope of this comparison study, related work is currently being performed by researchers with secondary gravity wave resolving models.

Regarding the reviewer's comment on balanced flow, for completeness we should mention that, under normal conditions, the usual effect of westward gravity wave drag on the residual circulation is indeed typically balanced by the Coriolis force of a wintertime poleward meridional circulation. Thereby, the drag determines the mean temperatures in the polar stratosphere and mesosphere by adiabatic heating from downwelling, while the mean zonal wind adjusts according to thermal wind balance. These details are perhaps best reserved for a specific technical paper on the issue, such as Becker [2012].

**1.2** Specific comments**

**1. In the introduction, the authors introduce WACCM-X. A short description needs to be included here (Line 63).**

Agreed, thank you – a describing sentence has been added to this line. "WACCM-X is the eXtended version of the Whole Atmosphere Community Climate Model, a general circulation model based on Community Earth System Model (CESM) more details can be found in Section 2.2."

**2. The authors note that QBO10 and QBO30 are 'orthogonal' near line 134. They should provide an explanation of what they mean by this.**

We have removed the word 'orthogonal' and re-worded the sentence for clarity. New sentence reads: "These two heights are chosen firstly because the two time series do not correlate with each other [Chiodo et al., 2014], and because these two indices are often used together in atmospheric regression studies [Nair et al., 2013, Bojkov and Fioletov, 1995, Ramesh et al., 2020] to capture possible influences of the QBO."

**3. Near line 136, the authors say that the influence of the SAM is important for Antarctic winds. How do they know this is true or that the SAM does not influence non-Antarctic winds. Please rephrase.**

The SAM describes the north/south movement of the belt of westerly winds that encircle the Antarctic Peninsula, so by definition the SAM is important for the Antarctic surface winds. This sentence has been rephrased for clarity and now includes references on the SAM and its influences beyond the Antarctic continent.

The sentence now reads as follows: "The SAM is the dominant mode of atmospheric variability in the Southern Hemisphere [Marshall, 2003] and beyond the Antarctic, the SAM has been found to have influences on the climate in Australia, South Africa and other Southern Hemisphere countries [Gillett et al., 2006]. We speculate however, that as a Southern Hemispheric oscillation, the SAM is often excluded from linear regression studies of global models where regional oscillations are not considered [Ramesh et al., 2020, Gan et al., 2017]."

4. In the discussion of multicolinearity in section 3.1.1, VIFs are introduced without a reference. It is also not clear what R is (it too needs a reference). An example on which dependent

**and independent variables contributed to the calculation of a VIF would perhaps clarify this section.**

We have now included a reference [Nair et al., 2013] to the introduction of VIFs and included a reference for  $R^2$  [Montgomery, 2012].

We agree that the description of this section is unclear and so we have re-worded it. We have also included a new figure in the manuscript (Figure 1 in this response), to aid in the explanation of the VIFs.

| VIF values |        |      |      |      |      |      |      |      |      |      |      |        |     |
|------------|--------|------|------|------|------|------|------|------|------|------|------|--------|-----|
| F10.7      | - 1.05 | 1.07 | 1.15 | 1.13 | 1.18 | 1.17 | 1.21 | 1.29 | 1.12 | 1.12 | 1.05 | 1.08 - |     |
| ENSO       | - 1.15 | 1.12 | 1.05 | 1.06 | 1.10 | 1.10 | 1.16 | 1.44 | 1.44 | 1.37 | 1.47 | 1.30 - | 1.8 |
| QBO10      | - 1.60 | 1.34 | 1.14 | 1.06 | 1.19 | 1.31 | 1.49 | 1.15 | 1.07 | 1.07 | 1.48 | 1.62 - | 1.0 |
| QBO30      | - 1.55 | 1.31 | 1.23 | 1.05 | 1.11 | 1.12 | 1.13 | 1.21 | 1.39 | 1.32 | 1.51 | 1.55 - | 1.4 |
| SAM        | - 1.07 | 1.10 | 1.17 | 1.02 | 1.22 | 1.27 | 1.47 | 1.08 | 1.09 | 1.12 | 1.15 | 1.05 - | 1.2 |
|            | DJF    | JFM  | FMA  | MAM  | AMJ  | MJJ  | JJA  | JAS  | ASO  | SON  | OND  | NDJ    | 1   |

Figure 1: Variance Inflation Factors (VIFs) for each three month period and each independent variable. These values are calculated by extracting the given months from the 11-year time period and carrying out a regression of the corresponding independent variable against the other four. We take the R-squared value from that regression and calculate the VIF using Equation 1.

$$VIF = \frac{1}{(1 - R^2)},\tag{1}$$

5. The authors should note the presence of the following two Antarctic observation and analysis papers in the context of both trends and linkages to climatological indices:

French W.J.R., Mulligan F.J., Klekociuk A.R.(2020) Analysis of 24 years of mesopause region OH rotational temperature observations at Davis, Antarctica – Part 1: long-term trends, Atmospheric Chemistry and Physics 6379–6394;doi:10.5194/acp-20-6379-2020

French W.J.R., Klekociuk A.R., Mulligan F.J. (2020) Analysis of 24 years of mesopause region OH rotational temperature observations at Davis, Antarctica – Part 2: Evidence of a quasiquadrennial oscillation (QQO) in the polar mesosphere, Atmospheric Chemistry and Physics 8691–8708; doi:10.5194/acp-20-8691-2020

Thank you for bringing these two papers to our attention, we agree that they provide a useful addition to our paper. We have now referenced both within the manuscript.

6. The discussion of gravity wave tendencies in section 6 and near line 434 should include consideration of the relationship between GW tendency and the background wind. Wave breaking is affected by background wind conditions so a discussion of GW tendency needs to consider the existing relationships between climatological factors and the winds discussed earlier in the paper. The plot title on Fig 8a should include a word or symbol that denotes tendency. (It is misleading when read by itself as it is.)

Following comments from two reviews about the zonal mean zonal gravity wave tendencies presented in this paper and how the zonal mean may not be representative of the local gravity wave tendencies, we have decided to remove the section on gravity wave tendencies. We thank the review for this particular comment but note that it is now redundant. 7. Near line 292 (first paragraph of section 5.2), the zonal 95-100km ENSO significant zone (fig 6c) seems to extend into June.

Thank you for noticing this - the text has been amended to "May-June".

**8. Near line 349 – The authors should note that the 10 deg latitude difference between Scott Base and Rothera could affect comparisons.**

Thank you, this is now noted in this paragraph

**9. Sentence starting L 443. At what height does the effect on GW forcing seen by Li et al (2016) occur?**

This is found at heights of 70-100 km. We have now included this in the paper.

**10. L35 'constrain' to 'constraining'**

Agreed and changed as requested.

**11. L64 move parenthesis after '1992' to after '(2009)'**

Agreed and changed as requested

**12. L94 Suggest delete 'In this section we discuss'**

Agreed and changed as requested

**13. L130 Suggest small 's' on 'Solar' and follow by a comma**

The comma has been added and all occurrences of 'Solar' have been replaced by 'solar' (where grammatically correct).

**14. L 182 replace 'to test' with 'which'**

Line 182 originally read "To test for auto-correlation, we use the Durbin Watson (DW) test, further details of the test can be found in...". We believe this comment intended to say replace '**the** test' with 'which'. We have made this change and the sentence now reads "To test for auto-correlation, we use the Durbin Watson (DW) test, further details of which can be found in..."

**15. L212 Start a new sentence after 'Figure 3'**

Agreed and changed as requested

**16. L282 Delete last 's' in 'westwards'**

Agreed and changed as requested

**17. L293 delete 'a' before 'south'**

Agreed and changed as requested

**18. L294 '... WACCM-X in either component'? This is now what I see. Please check wording.**

The wording here has been corrected.

**19. L333 Do you mean QBO30 in this line?**

Yes, thank you for spotting, changes made

**20. L391 insert 'by' after 'paper'**

Agreed and changed as requested

**21. L406 Suggest replace 'Although' with 'However'**

Agreed and changed as requested

**22. L466 replace 'affects' with 'effects'**

Agreed and changed as requested

**23. L473 End sentence after 'all heights'**

Agreed and changed as requested

**2 Reviewer 2**

**2.1 Main comments**

1. The data set and analysis of long-term observation over the Antarctic station and corresponding WACCM-X data are rare and valuable in understanding MLT dynamics and interannual variability. Results shown in '4. Results: The winds in radar observations and WACCM-X' is overall fine. However, the linear regression analysis results shown in '5. Results: Linear regression analysis' need more careful consideration and evaluation.

Thank you for this advice, we have addressed the main issues highlighted by this comment below.

2. My main concern is the number of samples in linear regression analysis. The sample for linear regression analysis is three-month window times 11 years, i.e. 33 samples (data points) for each time and height bin. The estimated coefficient ( $\beta$ ) is as many as six, as shown in Equation (1). I suppose this number of coefficient is too many for 11 years of data.

The reviewer is correct that more samples would be beneficial for the regression analysis in this paper. However, long term geophysical datasets of the MLT winds in Antarctica are very limited so we are restricted by the length of data set available. We argue that a regression with this number of observations and independent variables remains valid. For each regression we have 33 observations and 5 independent variables (F10.7, ENSO, QBO10, QBO30 and SAM). Hair [2006] states that "the minimum ratio of observations to variables is 5:1". For the analysis we perform in this paper, our ratio is 6.6:1. This is above the minimum ratio required and the regression with this number of samples is valid. However, the reviewer is correct in that we would ideally have more observations to feed into the model.

We have now changed the text to highlight this in the manuscript and have included the following sentences: "A final caveat of this method surrounds the number of samples used in this regression analysis. Whilst our ratio of observations to independent variables is valid for linear regression, it is on the low end; this could lead to the regression model being too specific to this time period and not generalisable to other solar cycle periods for example." 3. The variance in these 33 samples is not only due to the interannual (or year-to-year) variability, but also intra-seasonal (or month-to-month) variability, because variation in threemonth window is also included. Statistical significance of each regression coefficient must be overestimated because the interannual variability is not included in three sample in the same year window (of three month). Data number of 33 are not number of independent samples in the sense of interannual variation, and therefore freedom of 27 is overestimated.

This is an interesting point and we thank the reviewer for bringing this to our attention. This limitation of our method has now been highlighted in the linear regression methods section, we have included the following text; "Our use of three month windows (to triple the number of datapoints in the regression) means that the variance in the samples is not only due to interannual variability but will also include some intra-seasonal variability. However, the analysis still allows us to investigate the linear influences of atmospheric and solar oscillations on the winds in the Antarctic MLT and gain useful insights into potential drivers of variability." We have also edited the text to highlight this limitation again in the discussion section.

The sentence stating the degrees of freedom has been removed.

**4. Multicollinearity (3.1.1) check by VIF should be carefully evaluated because under the presence of both intra-seasonal and interannual variability VIF should be underestimated.**

We do not fully understand this comment. We interpret it to mean that careful application of the VIF calculation is needed, i.e. not just calculating the VIFs of the full timeseries of the independent variables, rather splitting them up into the three month windows (that are used in the regression) and calculating the VIFs on that. We note that we do the latter in this work. We agree that this is not entirely clear here so we have simplified and re-phrased the paragraph. We have also included a new figure (Figure 1 in this text - see response to reviewer comment 1.2.4), intended to help illustrate the calculation of the VIFs.

**5. Also, if the autocorrelation in 3.1.2 is calculated for all 11-year time-series, this will smooth out different auto-correlation function at different season (or three-month window) so the value should be underestimated. Effect of such seasonal smoothing (or averaging) should affect DW test, and this should be checked.**

The auto-correlation in Section 3.1.2 is calculated for the individual three month regression models rather than the full 11-year time-series. We believe that the issue the reviewer is concerned about with seasonal smoothing isn't a problem for our application of the DW test. We have clarified the application of the DW test in this section to remedy this.

**6. Therefore, my recommendation is that authors clarify the above questions before publishing the results of Chapter 5, 6 and 7.2.**

**2.2 Specific comments**

1. 'These zonal gravity-wave tendencies in WACCM-X were found to be noisy when examined over the meteor collecting region. We therefore calculated tendencies as zonal-means in a band of 300 km latitudinal width, centred over the latitude of Rothera.' This assumes that the gravity wave drag is uniform at all longitude if monthly averaged. Is this an appropriate assumption? It is known that Andes and Antarctic Peninsula is the region of strong gravity wave generation. I do not think the zonally uniform assumption is correct.

We have now removed the gravity wave tendencies section from the paper.

2. L 255 'In (a), it can be seen that the interdecile range of the zonal wind maximises over the summer, when the zonal wind reversal occurs, due to the considerable variability in the strength and timing of this reversal.' Please check whether this difference of the interdecile

**range could be due to the difference of zonal wind magnitude between summer and winter. If the fluctuation is a certain % of the amplitude, this also cause the variation of interdecile range measured by m/s.**

Please see the figure below (Figure 2) presenting the interdecile range divided by the monthly mean wind of each month. The stand out feature of this is that the ratio becomes large during some months. This is especially evident in August in the meridional direction for WACCM-X at 85 km. This is because the wind there is very close to zero, please see Figure 3, and so the fractional interdecile range is poorly defined. In regard to the reviewers question, the amplitude of the zonal wind in summer is generally higher than in winter (Figure 3a), as is the interdecile range than winter. It does appear that the interannual fluctuation is a percentage of the amplitude; however this is hard to interpret due to the wind being small during some months.

Interdecile Range / monthly mean wind

Figure 2: Interdecile range divided by monthly mean

**Monthly mean wind**

Figure 3: Monthly mean line plot

3. L 268-9 'Hatched regions show where the relationship is statistically significant at the 90% level, using the Student's t-test.' I am suspicious about this, considering the point described as 'main concern' above.

Please see our response to comments 2.1.2 and 2.1.3

**3 Community comment - Karanam Ramesh**

**3.1 Main comments**

1. I have major concerns in the present form of this work. The authors did not explain any strong reason why they considered only the above mentioned forcings in their regression analysis? As the winds are related to temperature gradients and they are in close agreement with the thermal wind balance, the climate index for temperature variability or its driving factor like CO2 or O3 has not been considered.

The main reason for not including other forcings in our analysis is due to the restriction on the number of independent variables that we can use. This is because our observational dataset is limited and we need to avoid overfitting our regression model. Long term observations of the wind in the MLT are very scarce, particularly in Antarctica. Please see our response to comment 2.1.2 where we cover this limitation in more detail.  $CO_2$  was specifically avoided due to its strong correlation with time and hence the solar cycle over the period 2005-2015. It would not be a beneficial regressor. Similarly, ENSO shows some correlation with total column ozone so we have not included ozone.

**2. The wind anomalies (mean removed) are used in the regression analysis, however it is not clearly stated whether the climate indices are used as their anomalies (mean removed). If not done so, the mean removed anomalies of the predictors must be used in the regression analysis.**

We are unclear on whether this comment means a) mean removed i.e. index time series minus mean value of entire time series (mean removed) or b) index anomalies i.e. deviation of each month from the climatological mean (in the same way that wind anomalies are defined in the text). To clarify, the indices are used as defined and presented in Figure 6 and in the manuscript. F10.7, ENSO, QBO10 and QBO30 are not pre-processed but the seasonality is removed from SAM i.e. we regress against the SAM anomaly. We agree that this was not clear in the text so has now been emphasised. We respond to the comment below for both (a) and (b) interpretations of this comment.

a) Use mean removed indices in the regression analysis - We have explored the use of mean removed indices in the regression analysis and this change makes no difference in the results. The mean removal simply shifts the indices but as the results are presented as wind changes in m/s per solar flux units (e.g. for the solar coefficient results), the constant shift in the indices make no difference. For this reason we have not changed the analysis to include this.

**b)** Use index anomalies for the regression analysis - We believe it is not necessary to take the anomalies of the indices in order for the regression to be valid. This is because F10.7, ENSO, QBO10 and QBO30 have no seasonal cycle (i.e. repeating oscillations over 12 months). Therefore using the mean removed value does not make physical sense. The SAM index however does exhibit a seasonal cycle and we do consider the SAM anomaly for the regression.

In order to validate our understanding of this, we have re-run the regression using the anomalies from all the indices. In Figures 4 and 5, panel (a) shows our original analysis with the regression with only the SAM anomaly, panel (b) shows all indices as anomalies. Please note that panel (a) differs subtly from those plots previously presented previously in the manuscript as we have changed from monthly medians to monthly means and reduced the significance level hatching to the 95% level (please see our responses to comments 3.2.2 and 3.2.34.). The differences between panel (a) and panel (b) in both zonal and meridional directions

are minimal, we believe this is because the mean removal of the indices doesn't change the general shape and merely shifts/distorts the indices slightly (please see lineplots below and note the difference y scales -Figure 6). Therefore we disagree that this step is necessary.